



# A Lagrangian analysis of upper-tropospheric anticyclones associated with heat waves in Europe

Philipp Zschenderlein[1], Stephan Pfahl[2], Heini Wernli[3], and Andreas H. Fink[1]

[1]Institute of Meteorology and Climate Research, Karlsruhe Institute of Technology, Wolfgang-Gaede-Str.1, 76131 Karlsruhe, Germany
[2]Institute of Meteorology, Freie Universität Berlin, Carl-Heinrich-Becker Weg 6-10, 12165 Berlin, Germany
[3]Institute of Atmospheric and Climate Science, ETH Zurich, Universitässtr. 16, 8093 Zurich, Switzerland

**Correspondence:** Philipp Zschenderlein (philipp.zschenderlein@kit.edu)

**Abstract.** This study presents a Lagrangian analysis of upper-tropospheric anticyclones that are connected to surface heat waves in different European regions for the period 1979 to 2016. In order to elucidate the formation of these anticyclones and the role of diabatic processes, we trace air parcels backwards from the upper-tropospheric anticyclones and quantify the diabatic heating in these air parcels. Around 25-45% of the air parcels are diabatically heated during the last three days prior to their arrival in the upper-tropospheric anticyclones and this amount increases to 35-50% for the last seven days. The influence of diabatic heating is larger for heat wave-related anticyclones in northern Europe and western Russia and smaller in southern Europe. Interestingly, the diabatic heating occurs in two geographically separated air streams. Three days prior to arrival, one heating branch (western branch) is located above the western North Atlantic and the other heating branch (eastern branch) is located to the southwest of the target upper-tropospheric anticyclone. The diabatic heating in the western branch is related to warm conveyor belts in North Atlantic cyclones upstream of the evolving upper-level ridge. In contrast, the eastern branch is diabatically heated by convection, as indicated by elevated mixed-layer convective available potential energy along the western side of the matured upper-level ridge. Most European regions are influenced by both branches, whereas western Russia is predominantly affected by the eastern branch. The western branch predominantly affects the formation of the upper-tropospheric anticyclone, and therefore of the heat wave, whereas the eastern branch is more active during its maintenance. For long-lasting heat waves, the western branch regenerates. The results from this study show that the dynamical processes leading to heat waves may be sensitive to small-scale microphysical and convective processes, whose accurate representation in models is thus supposed to be crucial for heat wave predictions on weather and climate time scales.

## 1 Introduction

Among various kinds of natural hazards, temperature extremes and especially heat waves during summer impose large impacts particularly on human health (Horton et al., 2016; Watts et al., 2018). Anthropogenic climate change has already increased the number of heat wave days during the last decades (Perkins et al., 2012), which is in line with an overall global-scale temperature increase (Horton et al., 2015). However, the warming is not spatially uniform (Field et al., 2014) and some regions, e.g. Europe, encounter changes in the frequency, persistence and maximum duration of regional circulation patterns associated with extreme



temperatures (Horton et al., 2015). It is therefore crucial to understand the processes that lead to the formation and maintenance

of these circulation patterns.

    Recently, Zschenderlein et al. (2019) provided an analysis of European heat waves in the time period 1979 to 2016, identified as regions with temperature anomalies exceeding both the 90th percentile of daily maximum temperatures and the 25th percentile of annual maximum temperatures for at least three days. In all subregions considered, from the Iberian Peninsula to western Russia, these heat waves were associated with either an upper-tropospheric ridge or a blocking flow pattern. Several earlier studies emphasised that heat waves in the midlatitudes are typically co-located with atmospheric blocking (Carril

et al., 2008; Pfahl and Wernli, 2012; Stefanon et al., 2012; Pfahl, 2014; Tomczyk and Bednorz, 2019). Heat waves in Southern and Central Europe are often caused by intense subtropical ridges extending to Southern Europe (Sousa et al., 2018) or by a displacement of a North Atlantic subtropical high to Central Europe (Garcia-Herrera et al., 2010). Both blockings and intense ridges are associated with anticyclonic flow anomalies in the upper troposphere, and these anticyclones are essential

for the persistence of the events and for the strong downwelling associated with intense adiabatic warming of the air parcels (Zschenderlein et al., 2019). Upper-tropospheric anticyclones can therefore be regarded as an essential dynamic precursor for the formation of surface heat waves. As a continuation of Zschenderlein et al. (2019), we here aim to investigate the formation of these anticyclones in a Lagrangian and potential vorticity (PV) framework.

    Both blockings and subtropical highs are associated with negative PV anomalies in the upper troposphere (Schwierz et al.,

2004). These anomalies are the result of isentropic advection of low-PV air or cross-isentropic transport of low-PV air along moist ascending air streams. The isentropic advection of low-PV air corresponds to (i) the mechanism introduced by Yamazaki and Itoh (2013), in which blocking is maintained by the absorption of synoptic-scale anticyclones or (ii) the quasi-adiabatic transport of air from lower latitudes, often ahead of extratropical cyclones (e.g. Colucci, 1985). Pfahl et al. (2015) and Steinfeld and Pfahl (2019) investigated, in a Lagrangian framework, the influence of diabatic heating on the formation and maintenance

of blocking. Up to 45% of the air masses in northern hemispheric blocks experience latent heating by more than 2 K during the three days prior to their arrival in the block, and this percentage increases up to 70% when considering a seven-day period (Pfahl et al., 2015). The contribution of latent heating to the formation and maintenance of blocking is not uniform. Latent heating is more important for the onset than for the maintenance of the block (Pfahl et al., 2015). And in northern hemispheric winter, the contribution of latent heating is much larger for blocks over the oceans than for continental blocks, while in summer

also continental blocks are substantially affected by latent heating (Steinfeld and Pfahl, 2019).

    Latent heating due to condensation of water vapour is not only restricted to the formation of blocking, it generally influences the upper-level ridge building and amplification (e.g. Pomroy and Thorpe, 2000; Grams et al., 2011). In the midlatitudes, synoptic-scale latent heating occurs within moist ascending air streams from the lower to the upper troposphere, so-called warm conveyor belts (WCBs) (Green et al., 1966; Harrold, 1973; Browning et al., 1973). The outflow of the WCB produces

negative PV anomalies at the level of the midlatitude jetstream and is therefore a key process for the upper-level ridge building (Madonna et al., 2014).

    Only very few studies so far specifically investigated the role of latent heating for the formation of upper-tropospheric anticyclones related to heat waves in summer. Quinting and Reeder (2017) analysed trajectories reaching the lower and upper





troposphere during heat waves over southeastern Australia. They emphasised the influence of cloud-diabatic processes over a
baroclinic zone to the south of the Australian continent on the formation of upper-tropospheric anticyclones.

This study therefore focuses on the role of diabatic heating for the formation and maintenance of upper-tropospheric anticyclones associated with heat waves in Europe. We apply an impact-oriented perspective, meaning that we study a particularly impact-related type of upper-tropospheric flow anomalies. The following questions are addressed:

(1) What are typical source regions of low-PV air masses that constitute the upper-tropospheric anticyclones associated with
European summer heat waves?

(2) Are there inter-regional differences in the contribution of diabatic heating to the formation of these anticyclones?

(3) Where and in which synoptic environment does the diabatic heating occur in airflows entering the anticyclones?

(4) Are there differences in the relevance of diabatic heating during the formation and maintenance of the anticyclones?

Section 2 provides an overview of the data and methods employed in this study. The results section 3 starts with a discussion
of the origin of the air parcels arriving in the upper-tropospheric anticyclones followed by a comparison of different regions in Europe. Subsequently, the locations of strong diabatic heating and their synoptic environments are presented. The results section closes with a comparison of the formation and maintenance of upper-tropospheric anticyclones. In section 4, a summary of the main findings and avenues for further research are presented.

## 2   Data and Methods

This section first describes the identification of upper-tropospheric anticyclones and their connection to the heat waves at the surface. Secondly, the calculation of the trajectories and the identification of diabatic processes are outlined. If not noted otherwise, all analyses are based on the ERA-Interim reanalysis of the European Centre for Medium-Range Weather Forecasts (Dee et al., 2011) on a $1° \times 1°$ longitude-latitude grid. To be consistent with Zschenderlein et al. (2019), we use the period between 1979 and 2016.

### 2.1   Identification of upper-tropospheric anticyclones

We aim to assign the surface heat waves in the six European regions (dashed boxes in Fig. 1b) used in Zschenderlein et al. (2019) to upper-tropospheric anticyclones. For defining upper-tropospheric anticyclones, we use a PV-approach introduced by Schwierz et al. (2004) that is based on the anomaly of the instantaneous, vertically averaged PV between 500 and 150 hPa with respect to the monthly climatology. To be identified as an upper-tropospheric anticyclone, the PV anomaly at a grid point
must fall below $-0.7\,\mathrm{PVU}$ ($1\,\mathrm{PVU} = 10^{-6}\,\mathrm{K\,kg^{-1}\,m^2\,s^{-1}}$). Pfahl and Wernli (2012) used this threshold for the definition of weak blocking and demonstrated that the link between weak blocking and northern hemispheric warm temperature extremes is particularly robust. We therefore choose the $-0.7\,\mathrm{PVU}$ threshold for defining upper-tropospheric anticyclones.



In a second step, we assign the upper-tropospheric anticyclone to the respective region. Exemplarily for Central Europe, Fig. 1a depicts a composite of the vertically averaged PV anomaly for all heat wave days. The composite shows a negative
upper-tropospheric PV anomaly with small standard deviations over Central Europe. In order to study the formation of the corresponding anticyclones, we define a rectangular box enclosing the −0.5 PVU contour line in the composite (black solid box in Fig. 1a) and assign all upper-tropospheric negative PV anomalies in this box to heat waves in Central Europe. The respective boxes for the other regions are shown in Fig. 1b. All grid points with PV anomalies below −0.7 PVU in the respective box during heat wave days in the corresponding region (dashed boxes in Fig. 1b) are identified as upper-tropospheric anticyclones.

## 2.2  Backward trajectories

Seven-day backward trajectories, driven by three-dimensional ERA-Interim wind fields on 60 vertical model levels, are calculated at each six-hourly time step with LAGRANTO (Sprenger and Wernli, 2015) for every heat wave day. Trajectories are initialised in the upper-tropospheric anticyclone and started from an equidistant grid ($\Delta x$=100 km horizontally) and vertically between 500 and 150 hPa every 50 hPa with the additional criterion that the PV at the respective level must be less than 1 PVU.
The latter excludes starting points in the stratosphere, similar to Steinfeld and Pfahl (2019). Physical parameters traced along the trajectories include temperature and potential temperature. The total number of trajectories is between 700,000 for Greece/ Italy and nearly 2,000,000 for Scandinavia.

In order to quantify diabatic processes along the trajectories, we evaluate whether diabatic heating or cooling dominates. For that, we calculate the highest ($\theta_{max}$) and lowest potential temperature ($\theta_{min}$) along the backward trajectories over a three or
seven day period. Diabatic heating is calculated as the difference ($\Delta\theta$) between $\theta_{max}$ and the preceding, i.e. closer to the origin, potential temperature minimum, whereas diabatic cooling is quantified as the difference ($\Delta\theta$) between $\theta_{min}$ and the preceding potential temperature maximum. If the diabatic heating exceeds the absolute value of the diabatic cooling, the trajectory belongs to the heating branch and vice versa. If the magnitude of diabatic cooling and heating are equal, the trajectory will be sorted in the cooling branch. This approach is similar to Steinfeld and Pfahl (2019), but differs in the categorisation of the diabatic
heating and cooling branches.

Figure 2 shows an example for a three-day period: the backward trajectory in Fig. 2a experiences stronger diabatic heating (red arrow in Fig. 2a) than cooling (blue arrow in Fig. 2a) and therefore belongs to the heating branch, whereas in Fig. 2b the diabatic cooling dominates and the trajectory is consequently sorted in the cooling branch.

## 2.3  Feature composites

To explore in which synoptic environment the air parcels in the heating regime are diabatically heated, we create composites of various features centred around the location of maximum diabatic heating. We show PV at 330 K, wind at 800 hPa, mixed-layer convective available potential energy (ML CAPE) and convective and large-scale precipitation. Whereas convective precipitation in ERA-Interim comes from the parameterised shallow, mid-level and deep convection, large-scale, i.e. stratiform, precipitation denotes the contribution coming from the cloud scheme (Dee et al., 2011). Flow features, i.e. blocks, cyclones
and warm conveyor belts, are taken from Sprenger et al. (2017). In their climatology, weak atmospheric blocking is defined as





a region where the anomaly of vertically averaged PV between 500 and 150 hPa is lower than $-0.7$ PVU and persists for at least five days (Schwierz et al., 2004; Croci-Maspoli et al., 2007). Hence, temporal persistence is required in addition to our definition of upper-level anticyclones. The region affected by a cyclone is defined as the region within the outermost closed sea level pressure isoline surrounding one or several local sea level pressure minima (Wernli and Schwierz, 2006). Warm conveyor

belts are air parcel trajectories ascending more than 600 hPa in two days associated with a midlatitude cyclone (Madonna et al., 2014). A more detailed description of the three features is given in Sprenger et al. (2017). To assess, whether the occurrence of blocks, cyclones and WCBs is anomalous, we compare the frequencies of the three features during diabatic heating with their climatological frequencies. The anomaly is then defined as the difference between the observed frequency during heat wave days and the climatological frequency.

## 130  3  Results

### 3.1  Source regions of low-PV air masses

This section focuses on the origin of trajectories started from the upper-tropospheric anticyclones. To this end, density maps of trajectory locations at specific time steps are created, which show relative frequencies and are normalised such that the spatial integral over the whole distribution yields 100%. We only present the density maps for heat waves in Central Europe, Western

Russia and Greece/Italy because they exhibit the largest differences.

Three days prior to the arrival of the trajectories in the upper-tropospheric anticyclone over Central Europe, one part of the heating branch is located over the western North Atlantic and the other part over northwestern Africa in the middle and partly lower troposphere (Fig. 3a). The western North Atlantic is a typical source region of diabatically heated trajectories for the formation of atmospheric blocking, although the main source region in summer is shifted towards North America (Pfahl et al.,

2015). In the blocking study by Pfahl et al. (2015), most of the backward trajectories were initialised over the North Atlantic to the west of Central Europe, which explains the westward shift of the source regions of diabatically heated trajectories compared to our study. Additionally, the western North Atlantic is the entrance region of the summer storm tracks (Dong et al., 2013) and therefore a region prone to diabatic heating. The second major source region over northwestern Africa (Fig. 3a) is not known as a source region for air parcels influencing the formation of blocking but appears to be important for the formation

of summertime upper-tropospheric anticyclones in association with heat waves. Due to this separation of the heating branch into two distinct regions, trajectories in the heating branch located west and east of 30°W three days prior to the arrival in the upper-tropospheric anticyclone are analysed separately in the following and are hereafter denoted as the western and eastern heating branch, respectively.

Air parcels in the cooling branch related to upper-tropospheric anticyclones above Central European heat waves are located

in the upper troposphere at around 300-400 hPa and mostly above northwestern Africa, but also over the North Atlantic and already within the upper-level anticyclone area three days prior to their arrival (Fig. 3b). These air parcels are then transported northwards to Central Europe along the western flank of the ridge associated with the heat wave. Pfahl et al. (2015) showed that the majority of the air parcels not influenced by diabatic heating (comparable to our cooling branch) are, three days prior





to the arrival in the block, located to the east of the diabatically heated trajectories. This is also the case here when comparing
the location of the cooling with the western heating branch (Figs. 3a,b).

Seven days prior to the arrival of the air parcels in the heating branch to Central Europe, most of them are located above
North America and the western North Atlantic and to some extent above northwestern Africa. Compared to the three-day
period, air parcels are located at lower altitudes (Fig. 4a). Generally, air parcels in the subtropics over the North Atlantic and
Gulf of Mexico are located at lower altitudes compared to air parcels above the North American continent and towards the
East Pacific. Air parcels in the cooling branch are at similar pressure levels compared to the three-day period, but more widely
distributed compared to the heating branch with a maximum density above the North Atlantic (Fig. 4b). Similar to the three-day
timescale, the major part of the cooling branch is found east of the heating branch.

The density maps for air parcels in the heating branch reaching Western Russia exhibit two distinct differences compared to
Central Europe. Firstly, the source regions of the heating branch do not show two clearly separated geographical maxima on the
three-day timescale (Fig. 3c). In fact, the major part of this branch is located above the European continent and in the middle
troposphere. However, on the seven-day timescale, a pattern of two geographical maxima emerges with the highest densities
over the western North Atlantic and in the Mediterranean area (Fig. 4c). Secondly, more air parcels are already located in the
vicinity of the target upper-level anticyclone indicating that the diabatic heating can occur more locally. The overall pattern of
the cooling branch, however, does not reveal substantial differences compared to the pattern for Central Europe, although the
maximum densities are generally shifted to the east (Figs. 3d and 4d).

Air parcels in the heating branch reaching the upper-troposphere above Greece/ Italy predominantly originate from north-
western Africa during the last three days, in particular from the Atlas Mountains (Fig. 3e). Therefore, these anticyclones are
strongly influenced by the eastern heating branch, whereas on the seven-day timescale, most of the diabatically heated trajec-
tories originate from the western Atlantic and North America (Fig. 4e). The majority of the air parcels in the cooling branch
are located above the North Atlantic three and seven days prior to the heat wave, but some trajectories are located in the trop-
ics south of 20°N at around 200 hPa (Figs. 3f and 4f) - an area which is climatologically influenced by upper-level easterly
winds in summer (Fink et al., 2017). In this region and during this time of the year, organised convection in the from of huge
mesoscale convective systems occurs in the ITCZ (InterTropical Convergence Zone) over the West African monsoon region.
Their upper-level poleward outflow turns eastward to feed the subtropical jet over Northern Africa and the Mediterranean (cf.
Fig. 1 in Lafore et al., 2010).

## 3.2  Two diabatic regimes

We now compare the statistical distributions of the potential temperature changes in the heating and cooling branch. Changes
in potential temperature during the last three and seven days prior to reaching upper-tropospheric anticyclones over Central
Europe are shown as probability density distributions. For both the three- and seven-day period, the shape of the cooling branch
features a Gaussian normal distribution, whereas the heating branch is more skewed (Fig. 5a). This skewness increases for the
seven-day period, implying an overall higher magnitude of diabatic heating along the trajectories on this timescale. During the
last three days, about 29% of the trajectories are influenced by diabatic heating and, consequently, 71% belong to the cooling





branch (Fig. 5b). On the seven-day timescale, 42% of the trajectories are in the heating branch (Fig. 5b). Hence, diabatic heating along trajectories substantially influences the formation of upper-tropospheric anticyclones above Central Europe.

The majority of trajectories in the cooling branch slightly descend and are radiatively cooled in the free atmosphere, while most of the trajectories in the heating branch ascend (not shown). Overall, the diabatic heating is a more rapid process compared to the diabatic cooling (not shown). Therefore, the heating branch can be interpreted as a strongly cross-isentropic branch transporting low-PV air from the lower to the upper troposphere, whereas the cooling branch is a quasi-adiabatic process that advects low-PV air towards the upper-tropospheric anticyclone, in line with the analysis of Pfahl et al. (2015) and Steinfeld

and Pfahl (2019) for blocks.

    The cross-isentropic transport of low-PV air from the lower to the upper troposphere in the heating branch is stronger for western Russia. During the last three days, about 44% of the air parcels reaching upper-tropospheric anticyclones above western Russia are affected by the heating branch, which is the highest fraction among the different European regions (Fig. 5b). For Scandinavia and the British Isles, about 35% of the air parcels are influenced by diabatic heating, which is slightly more than

for Central Europe. The Mediterranean area, i.e. Greece/ Italy and the Iberian Peninsula, however, is less influenced by the heating branch with only about 25% of the trajectories in this branch on the three-day timescale. During the last seven days, the relevance of the heating branch increases for all regions (Fig. 5b). The highest influence of the heating branch (about 50%) is found for trajectories reaching upper-tropospheric anticyclones above the British Isles, Scandinavia and western Russia. Interestingly, the increase of the fraction of diabatically heated air parcels from the three- to the seven-day period is smallest

for western Russia, indicating that heat wave anticyclones in western Russia are less influenced by remote diabatic heating beyond three days prior to their arrival in the anticyclone.

    Comparing the fraction of diabatically heated air parcels contributing to the formation of atmospheric blocks (Pfahl et al., 2015) with our findings, we conclude that the fraction is lower in our study. This can be explained by three main reasons: firstly, weather systems that are associated with diabatic heating such as extratropical cyclones and warm conveyor belts are

climatologically less frequent during summer (Madonna et al., 2014). Secondly, Pfahl et al. (2015) defined blocking with a more pronounced negative PV anomaly, and because more intense negative PV anomalies are associated with stronger latent heating in WCBs (Madonna et al., 2014), the influence of diabatically heated trajectories is reduced in our study. Thirdly, the quantification of diabatic heating along trajectories of Pfahl et al. (2015) is slightly different, because they only quantified the contribution of diabatic heating to the formation of blocking and did not account for diabatic cooling.

**3.3   Two geographically separated heating branches**

In the remainder of this study, we further analyse the heating branches for heat wave anticyclones in three regions. We focus on Central Europe and Greece/ Italy, which are affected by the eastern and western heating branches (Fig. 3a,e), and on western Russia, which is affected predominantly by the eastern heating branch (Fig. 3c).

    The spatial distributions of the locations of maximum diabatic heating along the trajectories for the eastern and western

heating branches are shown in Fig. 6. These locations are defined as the geographical positions at the end of the maximum 6-h increase of potential temperature in the last three days prior to reaching the upper-level anticyclones. The western heating





branch associated with anticyclones above Central Europe accounts for 50% of the whole heating branch. Most of its diabatic heating occurs over the central North Atlantic between 40°-50°N and 20°-40°W (Fig. 6a). Air parcels in the eastern heating branch are diabatically heated over the continent in a similar latitude band (Fig. 6b). For western Russia, only 8% of the heated
trajectories are in the western branch and the strongest diabatic heating occurs over the North Atlantic, but also over Central Europe (Fig. 6c). The dominant heating branch reaching western Russia is the eastern heating branch (92%, Fig. 6d). Most of the diabatic heating in this branch occurs over the European continent and mostly in the target area between 50° and 60°N. For Greece/ Italy, 69 (31)% of the heated trajectories are assigned to the eastern (western) heating branch. Air parcels in the western heating branch experience diabatic heating over the western North Atlantic (Fig. 6e). Local maxima of diabatic heating
in the eastern heating branch occur above the Atlas Mountains and the Alps (Fig. 6f), suggesting the importance of orographic ascent for the formation of upper-tropospheric anticyclones in this region. Overall, most of the diabatic heating in the eastern heating branch occurs close to the target region, whereas the western heating branch is associated with more remote diabatic heating. Most of the trajectories are diabatically heated at around 400 hPa (not shown), indicating that the air parcels are mostly heated due to latent heat release in clouds, as opposed to surface fluxes.

Although the western and eastern heating branches are geographically separated, it may be possible that the maximum diabatic heating occurs at the same time before arrival in the upper-tropospheric anticyclone. Around 48 to 54 h prior to arrival, the western heating branch experiences the strongest diabatic heating (Fig. S1). On the contrary, trajectories in the eastern heating branch are strongly heated between 24 and 36 h prior to arrival (Fig. S1). Hence, air parcels in the western branch are heated earlier compared to the eastern branch.

To explore which synoptic systems lead to the ascent and latent heat release in the two different heating branches, we create composites of different fields and frequencies of blocks, cyclones and warm conveyor belts centred around the location of maximum diabatic heating. To emphasise the structure of the most pronounced heating, we only considered trajectories in the composites that are diabatically heated by more than 5 K during the last three days.

The composite for the air parcels within the western heating branch reaching Central Europe is presented in Fig. 7, the
composites for the other regions are qualitatively similar (not shown). The upper tropospheric circulation, represented by PV at 330 K, is characterised by a trough upstream of the maximum diabatic heating (Fig. 7a). At the surface, extratropical cyclones are frequently located west and north of the diabatic heating maximum. The position of the extratropical cyclones west of the heating maximum is slightly east of the upper-level PV trough, which corresponds to the canonical configuration of cyclogenesis at the leading edge of an upper-tropospheric trough. In the warm sector of these cyclones, where southwesterly
winds prevail (Fig. 7a), lifting occurs according to quasi-geostrophic forcing (Holton and Hakim, 2013). Hence, it is meaningful that warm conveyor belts are found centred around the diabatic heating maximum and downstream of the extratropical cyclones (Fig. 7a). These ascending air streams release latent heat and lead to an increase of potential temperature. Therefore, the western heating branch is often influenced by cyclones in the North Atlantic storm track and latent heating in their warm conveyor belts.

Above the diabatic heating maximum, an upper-level ridge evolves and blocking frequencies are enhanced downstream
(Fig. 7a). ML CAPE values are usually low in this branch. To the southwest of the diabatic heating maximum, ML CAPE values strongly increase due to climatologically higher sea surface temperatures in the western North Atlantic south of 30°N.





To assess, whether the occurrence of the three features in the North Atlantic region is anomalous for this time of the year, we compare the frequencies of the three features during the diabatic heating with their climatological frequencies. In general, the anomalies of all three features attain their highest values in the vicinity of or at the position of the diabatic heating maximum

(Fig. 7b). To the west and southwest of the diabatic heating maximum, the observed cyclone frequency is about 15 percentage points higher than the climatology, which is an increase by a factor of about 1.5. In contrast, the anomalies of the cyclone frequency to the north are smaller, although the observed frequency is similar (Fig. 7a). As a result of the enhanced cyclone occurrence, also the existence of warm conveyor belts is anomalously high (Fig. 7b). In accordance with the anomalously high cyclone frequency north and northwest of the diabatic heating maximum, the blocking frequency is anomalously low in this

region. Downstream of the diabatic heating maximum, the blocking frequencies are higher and the cyclone frequencies lower than climatologically expected.

Steinfeld and Pfahl (2019) performed a similar composite analysis for the latent heating associated with blocks and found a more pronounced upper-level ridge pattern due to similar reasons as discussed at the end of section 3.2. Overall, the latent heating in the warm conveyor belts of extratropical cyclones is important for the formation of both atmospheric blocks and

upper-tropospheric ridges associated with heat waves. Also Quinting and Reeder (2017) highlighted the role of cloud-diabatic processes and ascending air streams for upper-level anticyclones during heat waves in southeastern Australia.

After discussing the synoptic conditions of the western heating branch, we now focus on the conditions of the eastern heating branch. In this branch, the diabatic heating maximum is located below the western part of an upper-tropospheric anticyclone, which is much more pronounced compared to the western heating branch (Fig. 8a,b). In contrast, the frequency of

both cyclones and WCBs at the position of maximum diabatic heating is reduced (WCBs are not visible in Fig. 8a,b; they occur with frequencies of less than 3%). Hence, the driving mechanisms of the latent heating differs between the two branches. The circulation at 800 hPa is more anticyclonic and much weaker in the eastern compared to the western heating branch. The most substantial difference between the two heating branches is the enhanced ML CAPE in the eastern heating branch (Fig. 8a,b), indicating the potential for convection. The absolute values of ML CAPE are, however, not extremely high, which may indicate

that convection is efficiently depleting the ML CAPE. Additionally, according to ERA-Interim, most of the precipitation in the eastern heating branch is indeed convective (Fig. 8a; more clear for western Russia in Fig. 8b), whereas precipitation in the western heating branch is predominantly stratiform (Fig. 7a). Cloud top temperatures derived from infrared Satellite imagery are between $-5$ and $-9°$ C at the location of maximum diabatic heating (not shown). Hence, we assume that in the eastern branch latent heating is driven by mid-level convection or deep convection that reaches from lower into mid levels.

The anomalies underline the importance of the enhanced blocking frequencies and ML CAPE values for the eastern heating branch (Fig. 8c,d). Although the anomalies show also a small positive anomaly of cyclone frequencies (Fig. 8c,d), the absolute frequency (Fig. 8a,b) is lower compared to the western heating branch (Fig. 7a). Comparing the two regions, western Russia shows slightly higher anomalies of blocking frequencies and ML CAPE at the location of maximum diabatic heating (Fig. 8d). The eastern heating branch has not yet been discussed in the literature on the formation of European blocking, but it appears

to be relevant for the formation of upper-tropospheric anticyclones in association with heat waves in summer.





### 3.4 Diabatic heating during the life cycle of heat waves

Here, we investigate the life cycle of the upper-tropospheric anticyclones associated with heat waves, i.e. the temporal sequence of the occurrence of the different heating branches. The contributions of the eastern and western heating branches and their relative importance with respect to the whole heating branch is quantified as a function of the duration of the heat waves. We

concentrate on the results for Central Europe, because this region is equally affected by both branches. Due to the definition of the heat waves (cf. Zschenderlein et al., 2019), all events have a minimum duration of three days (Fig. 9). About 70 events have a duration of three days, but only two of them last 13 days. We therefore start with the discussion of the results for the heat waves with a duration up to six days and then elucidate the findings for the longer-lived heat waves. For the latter category, the results are likely less robust due to the small number of events.

During the onset of a heat wave, the western heating branch is of primary importance (Fig. 9). The formation of the upper-tropospheric anticyclone is therefore strongly affected by air masses that are diabatically heated in extratropical cyclones in the North Atlantic region. After the first two days of the heat waves, the eastern heating branch with air masses originating from northwestern Africa and heated diabatically due to convection below the western part of the ridge gains relevance (Fig. 9), thus supporting the maintenance of the upper-tropospheric anticyclone. The fraction of trajectories in the whole heating branch,

i.e. western and eastern heating branch together, with respect to all trajectories slightly increases during the maintenance of the upper-tropospheric anticyclone (black line in Fig. 9). Hence, the influence of latent heating increases during the life cycle of the events mainly due to an intensification of the eastern heating branch. At first sight, this result is contradictory to the findings of Pfahl et al. (2015) and Steinfeld and Pfahl (2019), who showed that the influence of latent heating reduces during the maintenance phase of atmospheric blocks. However, the heating relevant for atmospheric blocking mainly occurs

in trajectories similar to our western heating branch, and this branch looses relevance for the maintenance (up to day five) of upper-tropospheric anticyclones also here (Fig. 9).

Overall, the formation of upper-tropospheric anticyclones depends mainly on the latent heating within extratropical cyclones in the North Atlantic storm track, whereas the maintenance is related to air masses that are diabatically heated due to convection above western and central Europe. Although this pattern seems to be relevant for most of the heat waves, longer lasting heat

waves show a different behaviour.

The maintenance of heat waves beyond six days duration is more influenced by the western heating branch compared to the maintenance of shorter lasting heat waves (Fig. 9). Note that these longer lasting heat waves occur only rarely, therefore results are variable from case to case and less robust. However, it seems that the western heating branch revives and has a comparable influence as during the onset of the heat wave. We therefore hypothesize that long-lived upper-tropospheric anticyclones cannot

be sustained by the eastern heating branch alone. Rather, cyclones over the North Atlantic and the associated latent heat release are relevant to maintain the negative PV anomalies in the upper troposphere above the heat wave areas. In addition, the fraction of the heating branch related to all trajectories increases for longer-lasting heat waves up to nearly 50%.



## 4    Conclusions

In this study, we analysed the contribution of latent heating to the formation and maintenance of upper-tropospheric anticy-
clones associated with heat waves in different parts of Europe. Based on heat waves identified in Zschenderlein et al. (2019), we
calculated backward trajectories from the anticyclones and separated the trajectories according to their potential temperature
changes. The heating branch was further subdivided according to the location of the air parcels three days prior to the arrival in
the upper-tropospheric anticyclone into an eastern and western heating branch. In the introduction, we raised specific research
questions that we aim to summarise for Central Europe with the help of Fig. 10.

1. What are typical source regions of low-PV air masses that constitute the upper-tropospheric anticyclones associated with
   European summer heat waves?

For Central European heat wave anticyclones, mainly two source regions exist. Three days prior to reaching the upper-
tropospheric anticyclones, air parcels in the cooling branch are located in the upper troposphere southwest of the target region,
mainly distributed between Central Europe and the central North Atlantic, peaking over the northwest coast of Africa (Fig. 10,
label 1). Air parcels assigned to the eastern heating branch are located mainly between Central Europe and the northwest coast
of Africa in the mid- to lower troposphere (Fig. 10, label 3), while air parcels in the western heating branch culminate between
eastern North America and the western North Atlantic (Fig. 10, label 2) at similar altitudes.

2. Are there inter-regional differences in the contribution of diabatic heating to the formation of these anticyclones?

Around 25-45% (35-50%) of the air parcels are diabatically heated during the last three (seven) days prior to the arrival
in upper-tropospheric anticyclones. The influence of diabatic heating increases towards northern Europe and western Russia
and decreases towards southern Europe. While most regions in Europe are - with varying magnitude - influenced by both the
eastern and western heating branch, western Russia is only influenced by one diabatic heating branch. The contribution of
diabatic heating increases substantially on the seven-day timescale except for western Russia.

3. Where and in which synoptic environment does the diabatic heating occur in airflows entering the anticyclones?

For most regions in Europe, the diabatic heating occurs in two geographically separated moist ascending air streams. But the
air streams differ not only in location, also the processes responsible for the diabatic heating are different. The western heating
branch is influenced by an enhanced activity of extratropical cyclones and associated warm conveyor belts over the North
Atlantic. Diabatic heating in this branch is accompanied by stratiform precipitation, in contrast to the eastern heating branch,
where convective-scale precipitation dominates. The moist ascent in the latter branch occurs closer to the target anticyclone in
an environment of enhanced ML CAPE and is also aided by orographic lifting.

4. Are there differences in the relevance of diabatic heating during the formation and maintenance of the anticyclones?

The activity in the North Atlantic and the associated latent heat release in cyclones and warm conveyor belts are of primary
importance for the onset of the upper-tropospheric anticyclones connected to the heat waves. Their maintenance is affected



by the more local diabatic heating in the eastern heating branch. For longer lasting heat waves, the western heating branch

regenerates and becomes more relevant compared to days 3-5, implying that the ridge connected to the longer lasting heat wave cannot be sustained without the transport of low-PV air to the upper troposphere within extratropical cyclones.

One shortcoming of our approach is that our trajectory calculations are not able to resolve small-scale convective processes. Hence, we possibly underestimate the effect of convection especially in the eastern heating branch and therefore the associated

diabatic heating. Recently, Oertel et al. (2019) showed that embedded convection in warm conveyor belts can influence the synoptic-scale circulation and increase the isentropic PV gradient at upper-levels in addition to the slantwise WCB ascent. However, we assume that for our climatological analysis the source regions will not substantially change, because also the convective ascending parcels are located in the vicinity of the slantwise ascending WCB (Oertel et al., 2019) and we argue that convective parameterisation is tuned to capture the climatological bulk effects of deep convection on rainfall and latent

heat release. For the the eastern branch, especially in the Greece/ Italy case, the pathway of individual trajectories affected by deep convection over the Atlas Mountains and the Alps might be more uncertain due to the proximity of convection to the heat wave region. Weisheimer et al. (2011) noted that a revised formulation of the convective parameterisations in the ECMWF model improved the predictability of the 2003 European heat wave. Interestingly, air parcels arriving over Greece/ Italy can originate from the upper-level easterlies over West Africa (see section 3.1). Pante and Knippertz (2019) show that

explicit convection over West Africa improves forecast of upper-level fields over Europe at 5-8 days lead time. Thus, it would be interesting to calculate online, convection-permitting trajectories in high-resolution model simulations (e.g. Miltenberger et al., 2013) to study the impact of convection over the North African subtropics and over southern Europe on the formation of European heat waves.

Our results have relevant implications for both weather and climate dynamics. The processes discussed in our study need

to be correctly simulated in both state-of-the-art numerical weather prediction and climate models. Diabatic processes affect the life cycle of Rossby wave packages and a misrepresentation of these processes can lead to reduced predictability (Rodwell et al., 2013). Grams et al. (2018) showed that a misrepresented warm conveyor belt in an upstream trough led to misforecasts in the onset of blocking situations over Europe. Also Rodwell et al. (2013) pointed out that convective situations in eastern North America led to a forecast bust over Europe. When considering a higher moisture content in the lower troposphere in

a generally warmer world (Held and Soden, 2006), the latent heat release in cyclones or convective systems may increase. The stronger latent heat release stimulates the ascent of air streams that produce more significant negative PV anomalies in the upper troposphere (Madonna et al., 2014). Hence, model experiments quantifying the amplitude and the size of the upper-tropospheric anticyclones subject to a changing moisture content would be helpful to estimate the influence of global warming on the dynamics of European heat waves.

*Author contributions.* PZ carried out the analysis and SP, HW and AF gave important guidance during the project. PZ wrote the manuscript and all authors provided feedback on the manuscript.



*Competing interests.* The authors declare that they have no competing interest.

*Acknowledgements.* We thank ECMWF for providing access to ERA-Interim reanalysis data and Michael Sprenger and Lukas Papritz
for providing access to the blocking, cyclone and warm conveyor belt datasets and Daniel Steinfeld for providing access to the vertically
averaged PV dataset. We also thank Julian Quinting for inspiring discussions. The research leading to these results has been conducted
within the subproject C4: Coupling of planetary-scale Rossby-wave trains to local extremes in heat waves over Europe of the Transregional
Collaborative Research Center SFB/TRR 165 "Waves to Weather", funded by the German Research Foundation (DFG).



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



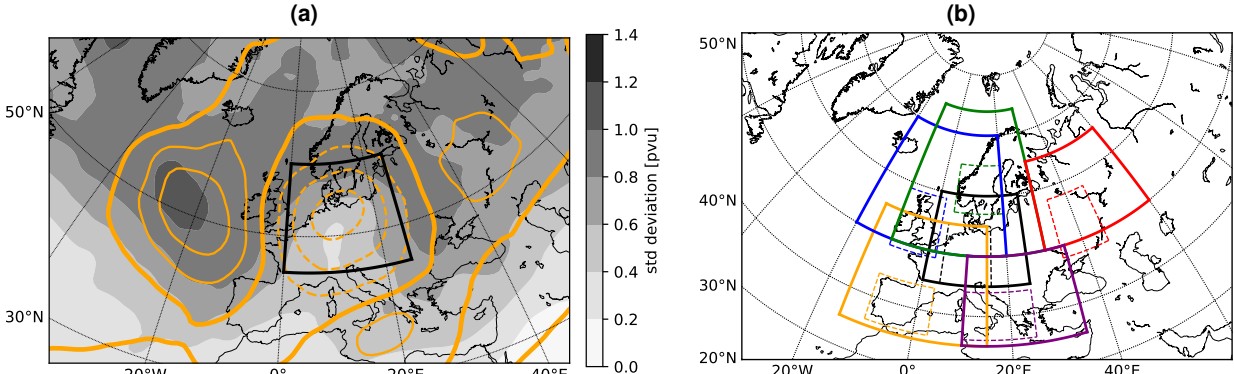

**Figure 1.** Identification of upper-tropospheric anticyclones. (a): Composite of the instantaneous, vertically averaged PV anomalies (VIPa) for all heat wave days in Central Europe. The contours show the mean of VIPa (in 0.25 PVU increments, 0 PVU line in bold, positive (negative) anomalies solid (dashed) and the shading shows the standard deviation of VIPa. (b): The solid boxes depict the regions where the upper-tropospheric PV anomalies are assigned to heat waves at the surface and the dashed boxes show the regions of the heat waves as defined in Zschenderlein et al. (2019): Scandinavia (green), western Russia (red), Greece/ Italy (purple), Iberian Peninsula (orange), Central Europe (black, also in (a)) and the British Isles (blue).





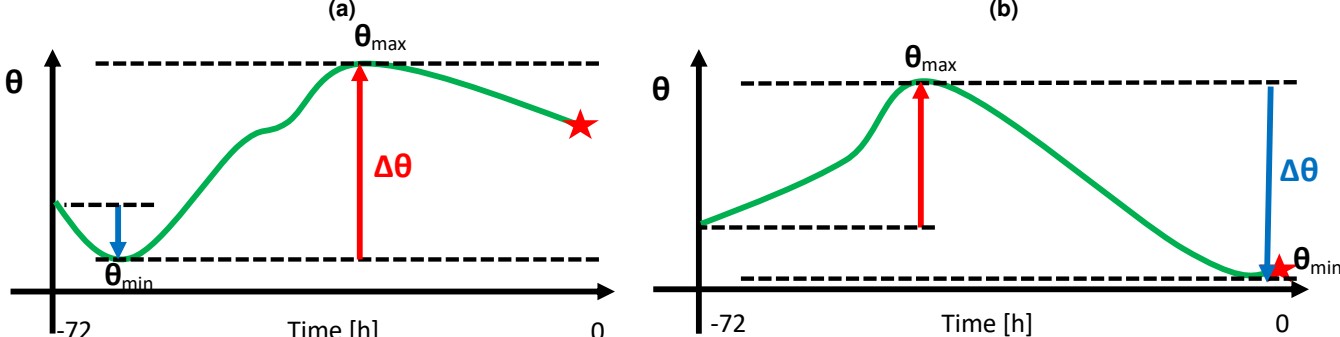

**Figure 2.** This schematic depicts the potential temperature change $\Delta\theta$ for a three-day period. The red star indicates the starting point of the backward trajectory. (a): Diabatic heating (red arrow) exceeds diabatic cooling (blue arrows). (b): diabatic cooling exceeds diabatic heating.




**Figure 3.** Spatial distribution of diabatically heated (left) and cooled (right) trajectories three days prior to arrival in the upper-tropospheric anticyclones for (a,b) Central Europe (CE), (c,d) western Russia (WR) and (e,f) Greece/ Italy (GI). The colours indicate the median pressure of air parcels and contours display the air parcel density (starting from 1‰ per $10^5$ km$^2$ in 2‰ increments). The dashed purple boxes represents the area in which upper-tropospheric anticyclones are associated with heat waves (cf. section 2.1 and Fig. 1).

**Figure 4.** Same as Fig. 3, but for the last seven days.





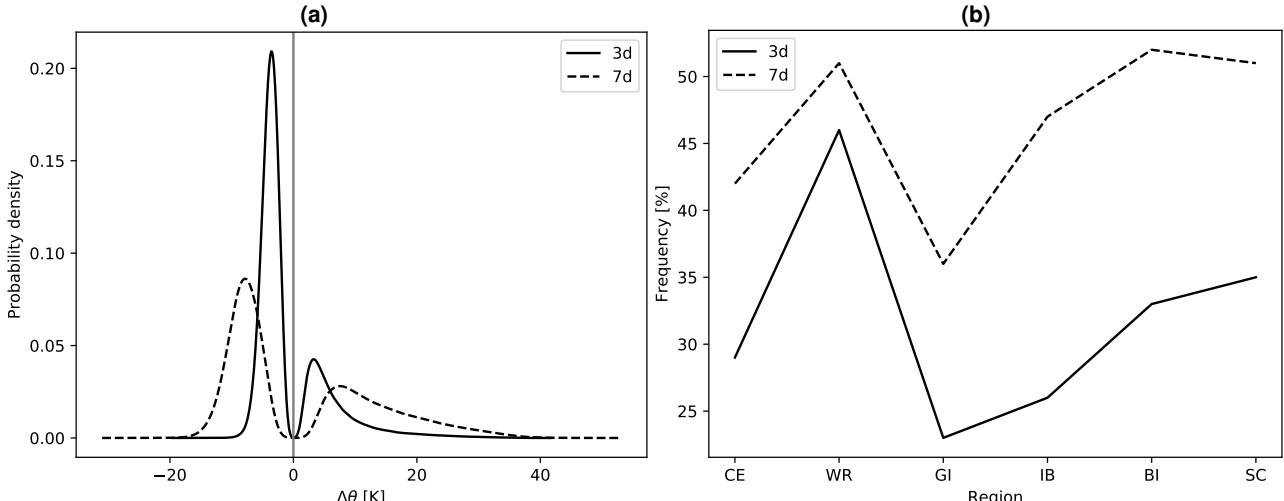

**Figure 5.** Diabatic processes in the heating and cooling branches three and seven days before reaching upper-tropospheric anticyclones. (a): Probability density distribution of the potential temperature changes for air parcels reaching Central European heat wave anticyclones. The grey line denotes the 0 K border separating the heating and cooling branch. (b): Fraction of diabatically heated trajectories for all regions (CE: Central Europe, WR: Western Russia, GI: Greece/ Italy, IB: Iberian Peninsula, BI: British Isles and SC: Scandinavia).





**Figure 6.** Geographic location of the maximum diabatic heating along trajectories for the western (left column) and eastern heating branch (right column) during the last three days prior to reaching upper-tropospheric anticyclones above Central Europe (CE) and western Russia (WR). The percentages in the orange boxes denote the fraction of the western/ eastern heating branch with respect to the whole heating branch.

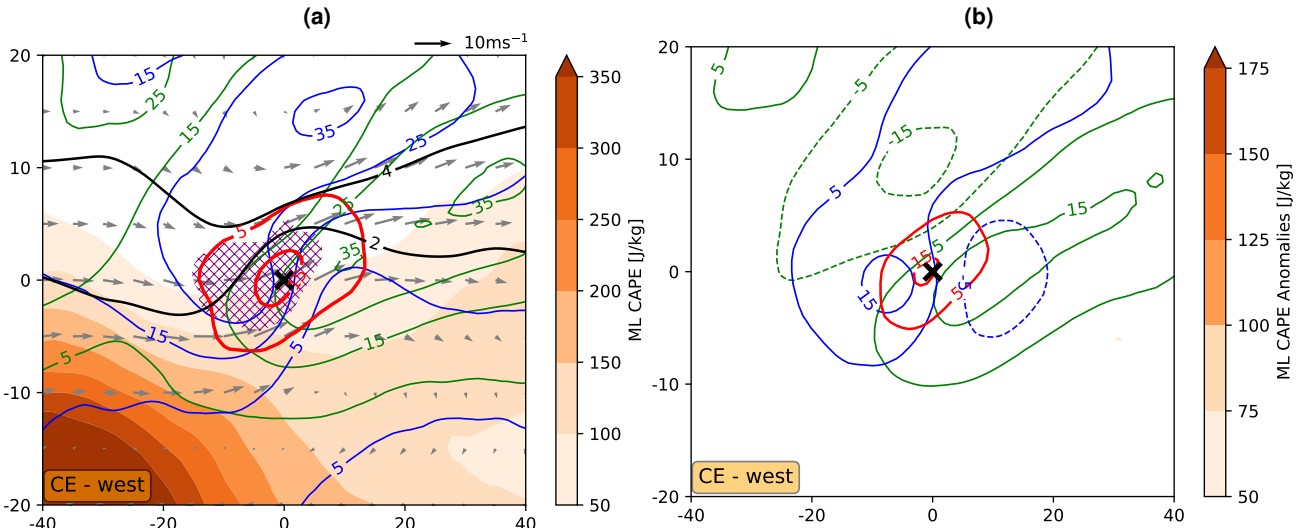

**Figure 7.** Composites centred around the position of maximum diabatic heating of the trajectories in the western heating branch reaching upper-tropospheric anticyclones above Central Europe. (a): Frequencies of extratropical cyclones (blue), blocks (green) and warm conveyor belts (red) starting from 5% in 10% increments. The orange shading shows the ML CAPE (in $J\,kg^{-1}$) and the arrows the wind at $800\,hPa$. Black contours indicate PV (2 and 4 PVU contours) at 330 K. The purple hatching marks the region where the stratiform precipitation exceeds the convective precipitation (only for areas with total precipitation $\geq 2mm/d$). (b) Anomalies of cyclone (blue), blocking (green) and warm conveyor belt frequency starting from 5 percentage points with 10 points increments. Orange shading shows ML CAPE anomalies (in $Jkg^{-1}$).

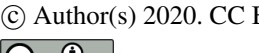

**Figure 8.** Same as Fig. 7, but for the eastern heating branch and Central Europe (a,c) and western Russia (b,d). The top row show the full fields and the bottom row the anomalies. The grey shading marks the area where the convective precipitation exceeds the stratiform precipitation.





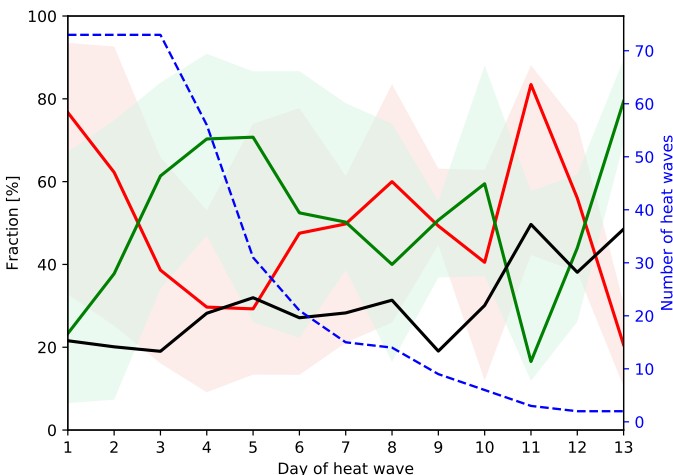

**Figure 9.** Latent heating during the life cycle of upper-tropospheric anticyclones connected to heat waves in Central Europe. The red (green) line shows the median contribution of the western (eastern) heating branch to the whole heating branch, the shading represents the range between the 25$^{th}$ and 75$^{th}$ percentile. The median fraction of the heating branch relative to all trajectories is represented by the black line and the number of heat waves is indicated by the blue dashed line.

**Figure 10.** Schematic illustrating the pathway of the three air streams contributing to the upper-tropospheric anticyclone (red cylinder) above the heat wave in Central Europe (black dashed line) during the last three days prior to arrival. Air stream 1 denotes the cooling branch and air streams 2 and 3 the western and eastern heating branches, respectively. Grey-marked lines at the surface illustrate the projections of the arrows (brighter colours indicate a higher altitude of the associated air stream). The bold black line represents the tropopause. The arrow of air stream 1 is wider because this branch is less spatially coherent compared to air streams 2 and 3.