# Peer review of "A Lagrangian analysis of upper-tropospheric anticyclones associated with heat waves in Europe"

_Weather and Climate Dynamics, 2019_

## Referee Comment (RC1) · Anonymous Referee #1 · 30 Jan 2020

In order to simplify the revision task, I organized my remarks taking into consideration the list of aspects suggested in the WDC review criteria, as follows: 1. Does the paper address relevant scientific questions within the scope of WCD? The manuscript addresses interesting questions, concerning synoptic conditions and processes leading to the occurrence of heat waves in Europe, which, regarding the global warming and increasing frequency of positive temperature extremes, is scientifically relevant and perfectly comprises the scope of the journal Weather and Climate Dynamics. 2. Does the paper present novel concepts, ideas, tools, or data? The concept of analyzing the role of diabatic heating for the formation and maintenance of upper-tropospheric anticyclones associated with heat waves, which was undertaken in the paper is novel

and it was pursued with adequate modern methods of a Langrarian analysis. Authors defined the backward trajectories of air parcels in the days prior to heat waves and quantified diabatic processes along the trajectories, which influenced formation of anticyclones. 3. Are substantial conclusions reached? Relevant, although surprising conclusions concerning the two source regions of air masses were obtained for heat waves in Central and Southern Europe. Described spatiotemporal variability of the diabatic processes influencing formation and conditions of anticyclones related to heat waves, seems to be one of the most important results. 4. Are the scientific methods and assumptions valid and clearly outlined? Data and Methods section is well organized and clearly written. Description of all calculations and research procedures are complete and precise; all methods are adequate to the anticipated results. 5. Are the results sufficient to support the interpretations and conclusions? Interpretations and conclusions in general well correspond to the obtained results, however, some conclusions concerning other European regions than the three analyzed in the study in details (Central Europe, western Russia and Greece/ Italy) seem to be weekly documented. Other regions are addressed only in single paragraphs and figures (Fig. 1b and 5b). I would suggest to consider removing regions IB, BI and SC from the analysis. 6. Is the description of experiments and calculations sufficiently complete and precise to allow their reproduction by fellow scientists (traceability of results)? Please, see point 4. 7. Do the authors give proper credit to related work and clearly indicate their own new/original contribution? Yes 8. Does the title clearly reflect the contents of the paper? The title is adequate to the content. 9. Does the abstract provide a concise and complete summary? Yes. 10. Is the overall presentation well structured and clear? Yes. 11. Is the language fluent and precise? Yes. 12. Are mathematical formulae, symbols, abbreviations, and units correctly defined and used? Yes. 13. Should any parts of the paper (text, formulae, figures, tables) be clarified, reduced, combined, or eliminated? Some clarifying of figures seems to be needed, namely: Fig. 1a – please adjust caption concerning PVU lines to the content of the map; it would be useful to put values on PVU isolines. Fig. 1b – I would suggest

to consider to delete Fig. 1b and eliminate from the analysis regions IB, BI and SC (please, see remarks in point 5). Fig. 5a – in my copy the difference between 3d and 7d line is not distinct enough. Fig. 5b – please, see comment in point 5. Fig. 7 and 8 – I would rather suggest to join the figures; please, note that captions are not complete (what does the black checked field mean?) Fig. 10 – Please, adjust the caption (I can't see the black dashed line in the picture). 14. Are the number and quality of references appropriate? The paper contains a reach list of references pertaining to both methods and comparable results. 15. Is the amount and quality of supplementary material appropriate? There is no supplementary material.

Please also note the supplement to this comment:
http://www.weather-clim-dynam-discuss.net/wcd-2019-17/wcd-2019-17-RC1-supplement.pdf

---

## Referee Comment (RC2) · Anonymous Referee #2 · 14 Feb 2020

Summary

This study follows Pfahl et al. (2015) and Steinfeld and Pfahl (2019) - most of the tools are used there – but this study targets heat-wave-associated upper-tropospheric anticyclones, which often can be weaker summertime continental anticyclones. Such target allows this study to find two heating branches: western (Atlantic, related to warm conveyor belt, stratiform precipitation) and eastern (continental, related to ML CAPE, convective precipitation, orographic lifting). The latter is not known in previous studies, potentially because analyses in previous studies (including the above two) might tend to be dominated by stronger oceanic blocking.

[Figure]

I find this study scientifically significant and methodologically sound. It may be well suited for publication if the presentation quality can be further improved and a couple of scientific comments below are addressed.

Major comments

1. (Line 281): As mentioned in line 358, trajectories do not resolve sub-grid scale convective processes. So, how is the eastern branch heated? Is it heated by the weaker stratiform precipitation?

2. I view the physical difference between the two heating branches is more about the heating mechanisms, less about where they are 3 days ago. Would it be cleaner to define the two branches based on their "locations of maximum diabatic heating", instead of where they are 3 days ago?

3. The novelty of this study against related work is not explicitly emphasized enough. In introduction, readers might want to know the deficiencies in related works that you will be solving, and in what way you might expect your results to differ from theirs.

As mentioned in my summary: I view this study is novel in targeting heat-wave-associated upper-tropospheric anticyclones. These anticyclones often collocate with heat waves [Roethlisberger et al. 2016, https://doi.org/10.1002/2016GL070944; Brunner et al. 2018, https://doi.org/10.1029/2018GL077837; Chan et al. 2019, https://doi.org/10.1029/2019GL083307], and are therefore continental anticyclones.

Line 57: Instead of throwing out all the key words, you can emphasize on heat wave anticyclone, saying that they often can be weaker summertime continental anticyclones and therefore may differ from global studies like Pfahl et al. (2015) and Steinfeld and Pfahl (2019), analyses in which might be dominated by more frequent oceanic blocking.

Line 60: Do you expect this study to be different and better than Quinting and Reeder (2017)? If so, please explicitly tell the difference.

Line 145: Again, lack of detailed studies of continental blocking could be the reason

why this eastern branch is not known before. Could emphasize on that.

Minor comments

You don't have to, but I personally find the naming of western/eastern branch not intuitive enough. Is there a better alternative?

The separation line of 30W is not repeatedly mentioned enough. Line 217: You might want to repeatedly remind readers that eastern means east of 30W and western means west of 30W, in this line and many other lines. Line 327: You might want to repeat in conclusions that 30W divides the two branches.

Line 9: "located southwest of the anticyclone" and "above western North Atlantic" are not mutually exclusive, consider saying "is located *over Africa/Europe* to the southwest..."

Line 22: Warming being "not spatially uniform" doesn't seem to connect well with the idea of changes in "regional circulation patterns".

Line 26: This paragraph can start with a better topic sentence, saying that heat waves are associated with either an upper-tropospheric ridge or a blocking flow pattern.

Line 27-28: In introduction, probably you don't need to include the fine details of methods in previous research.

Line 82: Would be good to exemplify upfront that for Central Europe, 72? heat waves lasted for at least 3 days are identified.

Line 87: You might want to explicitly mention that your definition of upper-tropospheric anticyclone requires no temporal persistence (this is implied in line 122).

Line 109: Might be good to be slightly clearer about the difference in method to Steinfeld and Pfahl (2019).

Line 233: Might be useful to show the figure for the pressure of maximum diabatic

heating.

Line 270: Is the idea of Quinting and Reeder (2017) more like warm conveyor belts in the western branch? Or more like the eastern branch? Could be more explicit.

Figure captions: Proofreading or copy-editing is needed. (Plurals, Capital letters, spaces, etc.)

Figure 7b caption line 6: Please note in caption that orange shading in 7b is not visible.

Figure 8 caption: Please note in caption that WCB is not visible.

Figure 8 purple hatching: Do you require total precipitation >= 2 mm/d? You might also want to remind readers that purple hatching in Fig. 8 is opposite to that in Fig. 7.

Technical corrections

Line 162: is found east of the *western* heating branch.

Line 224: over the *European* continent. . .

Line 236: *42* to 54 h

Line 296: "About 70" -> "72"?

Line 297: a duration of *at least* three days

Line 332: Are there *three* source regions instead of two?

Line 376: Do you mean Rossby wave *packets*?

Figure 3 caption line 4: boxes represents -> boxes represent

Figure 6 caption line 3: ... and Greece/ Italy (GI).

Figure 7 caption line 6: warm conveyor belts *(red)* frequency

Figure 8 caption line 2: grey shading -> purple hatching?

[Figure]

---

## Author Comment (AC1) · 20 Mar 2020

**Replies to the reviewers of "A Lagrangian analysis of upper-tropospheric anticyclones associated with heat waves in Europe" by Philipp Zschenderlein et al.**

We thank the reviewers for their interest and the time they spent to review our manuscript. The comments were very constructive and helped us to improve the quality of the manuscript. The comments of the reviewers are given in black, our replies in green colour.

**Review #1**

In order to simplify the revision task, I organized my remarks taking into consideration the list of aspects suggested in the WDC review criteria, as follows:

1. Does the paper address relevant scientific questions within the scope of WCD?
The manuscript addresses interesting questions, concerning synoptic conditions and processes leading to the occurrence of heat waves in Europe, which, regarding the global warming and increasing frequency of positive temperature extremes, is scientifically relevant and perfectly comprises the scope of the journal Weather and Climate Dynamics.

2. Does the paper present novel concepts, ideas, tools, or data?
The concept of analyzing the role of diabatic heating for the formation and maintenance of upper-tropospheric anticyclones associated with heat waves, which was undertaken in the paper is novel and it was pursued with adequate modern methods of a Langrarian analysis. Authors defined the backward trajectories of air parcels in the days prior to heat waves and quantified diabatic processes along the trajectories, which influenced formation of anticyclones.

3. Are substantial conclusions reached?
Relevant, although surprising conclusions concerning the two source regions of air masses were obtained for heat waves in Central and Southern Europe. Described spatiotemporal variability of the diabatic processes influencing formation and conditions of anticyclones related to heat waves, seems to be one of the most important results.

4. Are the scientific methods and assumptions valid and clearly outlined?
Data and Methods section is well organized and clearly written. Description of all calculations and research procedures are complete and precise; all methods are adequate to the anticipated results.

Replies to comments 1-4: We thank the referee for his/her comments.

5. Are the results sufficient to support the interpretations and conclusions?
Interpretations and conclusions in general well correspond to the obtained results, however, some conclusions concerning other European regions than the three analyzed in the study in details (Central Europe, western Russia and Greece/ Italy) seem to be weekly documented. Other regions are addressed only in single paragraphs and figures (Fig. 1b and 5b). I would suggest to consider removing regions IB, BI and SC from the analysis.

We discuss results for the regions IB, BI and SC in more detail in the revised version and added the respective figures (Figs. S1-S8) to the supplementary material. Explicitly, we added the following paragraphs:

> *"Three days prior to the arrival of the air parcels in the heating branch over the Iberian Peninsula and the British Isles, most of them are located above the western North Atlantic in the middle and lower troposphere, but also over northwestern Africa and Spain (Figs. S1a,c). For Scandinavia, air parcels are located over the western North Atlantic and southern/central Europe in nearly equal parts (Fig. S1e). On the seven-day time scale, air parcels of the heating branch are distributed between North America and the western Atlantic (Figs. S2a,c), although the dichotomy in the trajectory origin for Scandinavia still exists (Fig. S2e). The results for the cooling branches are qualitatively similar to the other regions (Figs. S1b,d and S2b,d,f), while for Scandinavia, a large fraction of diabatically cooled air parcels is already located in the target area three days prior to arrival (Fig. S1f)."*

Motivated by second reviewer, we have changed the terminology of the two heating branches. Instead of western branch, we use remote branch and instead of eastern branch, we name it nearby branch. This terminology is also used in our replies.

> *"The dominant remote branch reaching anticyclones above the Iberian Peninsula and the British Isles is diabatically heated above the central North Atlantic (Figs. S3a,c), similar to anticyclones over Central Europe. Scandinavia is slightly more influenced by the nearby branch (Figs. S3e,f) and air parcels in this branch are diabatically heated above central and western Europe (Fig. S3f)."*

> *"Trajectory-centred composites for the remote branch reaching anticyclones over western Russia, as well as for both heating branches arriving over the Iberian Peninsula, British Isles, Scandinavia and Greece/ Italy can be found in the supplementary material (Figs. S4-S8). Overall, the composites are qualitatively similar to the already discussed ones, especially for the remote branches (Figs. 4-8a,b), and only differ with respect to the magnitudes of ML CAPE in the nearby branch. ML CAPE values for Greece/ Italy are comparable to those for western Russia, albeit in a smaller area (Figs. S8c,d), but generally lower for trajectories of the nearby branch reaching anticyclones over Scandinavia (Figs. S7c,d) or the British Isles (Figs. S6c,d). In addition, the upper-level ridge of the nearby branch reaching Scandinavia (Fig. S7c) is more pronounced compared to Greece/ Italy (Fig. S8c). A similar difference in the magnitude of the upper-level ridge is found for the remote branches (Figs. S7-8a)."*

6. Is the description of experiments and calculations sufficiently complete and precise to allow their reproduction by fellow scientists (traceability of results)?
Please, see point 4.

7. Do the authors give proper credit to related work and clearly indicate their own new/original contribution?
Yes

8. Does the title clearly reflect the contents of the paper?
The title is adequate to the content.

9. Does the abstract provide a concise and complete summary?
Yes.

10. Is the overall presentation well structured and clear?
Yes.

11. Is the language fluent and precise?
Yes.

12. Are mathematical formulae, symbols, abbreviations, and units correctly defined and used?
Yes.

Replies to comments 6-12: We thank the referee for his/her comments.

13. Should any parts of the paper (text, formulae, figures, tables) be clarified, reduced, combined, or eliminated?
Some clarifying of figures seems to be needed, namely:
Fig. 1a – please adjust caption concerning PVU lines to the content of the map; it would be useful to put values on PVU isolines.

We added contour labels of the PVU lines and adapted the caption accordingly.

Fig. 1b – I would suggest to consider to delete Fig. 1b and eliminate from the analysis regions IB, BI and SC (please, see remarks in point 5).

See our reply to your comment 5.

Fig. 5a – in my copy the difference between 3d and 7d line is not distinct enough.

We adapted the line style for the 7d line and hope that they are now better discernible.

Fig. 5b – please, see comment in point 5.

See our reply to comment 5.

Fig. 7 and 8 – I would rather suggest to join the figures; please, note that captions are not complete (what does the black checked field mean?)

We think that two separate figures are more suitable, as they represent two branches with different mechanisms. Furthermore, with 6 panels, the images would be too small. We corrected the captions.

Fig. 10 – Please, adjust the caption (I can't see the black dashed line in the picture).

We meant the black dotted circle, maybe the term "black line" was misleading, we changed it to "black dashed circle".

14. Are the number and quality of references appropriate?

The paper contains a reach list of references pertaining to both methods and comparable results.

Thank you.

15. Is the amount and quality of supplementary material appropriate?
There is no supplementary material.

We uploaded various figures to the supplementary material.

**Review #2**

This study follows Pfahl et al. (2015) and Steinfeld and Pfahl (2019) - most of the tools are used there – but this study targets heat-wave-associated upper-tropospheric anticyclones, which often can be weaker summertime continental anticyclones. Such target allows this study to find two heating branches: western (Atlantic, related to warm conveyor belt, stratiform precipitation) and eastern (continental, related to ML CAPE, convective precipitation, orographic lifting). The latter is not known in previous studies, potentially because analyses in previous studies (including the above two) might tend to be dominated by stronger oceanic blocking.
I find this study scientifically significant and methodologically sound. It may be well suited for publication if the presentation quality can be further improved and a couple of scientific comments below are addressed.

We thank the referee for his/her comments and agree with the referee's assumption that previous studies tend to be dominated by stronger oceanic blocking, while our approach clearly emphasises summertime continental anticyclones.

Please note that we changed the terminology of the western and eastern branches. We now use remote and nearby branch, respectively.

Major comments
1. (Line 281): As mentioned in line 358, trajectories do not resolve sub-grid scale convective processes. So, how is the eastern branch heated? Is it heated by the weaker stratiform precipitation?

With our discussion in lines 358-373 we noted that sub-grid scale convective ascent is not resolved in ERA-Interim. In order to explicitly resolve sub-grid scale processes with trajectories, the only way would be to run a high-resolution model with explicit convection and calculate online trajectories. Since this is not feasible for a nearly 40-year climatology, we have to live with some limitations in the representation of convective processes. However, in ERA-Interim, the effect of convection is parameterized (cf. Dee et al., 2011, section 3.1.1) and the trajectories will capture the bulk effect of latent heating by

convection, even if they don't follow the rapid vertical ascent in convective updrafts. It is important to note that the composites (Figures 8 and 9) show remarkable differences concerning the feature frequencies and ML CAPE between the two branches. Also, the ratio between stratiform and convective precipitation differs. While stratiform precipitation dominates in the remote branch (Figure 8), convective precipitation dominates in the nearby branch, especially for western Russia (Figure 9). Hence, we assume that the precipitation in the remote (warm-conveyor belt) branch is predominantly heated by stratiform precipitation and the nearby branch (enhanced ML CAPE) by convective precipitation. Unfortunately, we are not able to identify whether parcels are heated by shallow-, mid-level or deep-convection. Since precipitation rates in the nearby branch are not very high in the composite, we assume that convection is mostly not very intense.

Since we explained this issue already in our discussion, we decided not to make any further amendments to the text.

2. I view the physical difference between the two heating branches is more about the heating mechanisms, less about where they are 3 days ago. Would it be cleaner to define the two branches based on their "locations of maximum diabatic heating", instead of where they are 3 days ago?

We performed a sensitivity study testing the new categorisation of the heating branches according to the "locations of maximum diabatic heating". Fig. R1 shows the location of maximum diabatic heating for heated trajectories reaching upper-tropospheric anticyclones over Central Europe. Some trajectories are heated above the North Atlantic and some over the European continent. At a first glance, it is difficult to find a clear longitude that separates the two heating branches, which is different to our approach, where locations three days prior to arrival are clearly separated (cf. Figs. 3a,c,e in the manuscript). Fig. R2 (left) shows a steady increase in the number of heated trajectories between 120°W and 18°W, followed by a sharper increase eastward of 18°W, which is somewhat more discernible for heated trajectories reaching anticyclones over Greece/ Italy (Fig. R2, right).

[Figure]

**Figure R1:** *Location of maximum diabatic heating for diabatically heated trajectories reaching upper-tropospheric anticyclones above Central Europe. Red colour shading shows trajectory counts per grid point.*

[Figure]

**Figure R2:** *Longitudinal distribution of diabatically heated trajectories reaching anticyclones over Central Europe (left) and Greece/ Italy (right).*

We have thus chosen 18°W as the longitude separating the remote and nearby branch. The resulting composites are qualitatively similar to our former approach (Fig. R3), although the magnitudes of the anomalies are slightly more pronounced, e.g. the WCB frequency is higher, and the area of convective precipitation is larger compared to our former approach, which is mainly due to the clearer geographical separation of the locations of maximum diabatic heating. Finally, we have looked at the temporal evolution of the two heating branches during the life cycle of the heat waves. The overall result is similar to the old approach (Fig. R4, left), although the relative difference between the remote and nearby branch during the onset is reduced. However, this result is highly sensitive to the definition of the longitude separating the two heating branches. When using 0°E as border, results for the life cycle are different (Fig. R4, right).

Overall, we conclude from this sensitivity analysis that pros and cons exist for both approaches. The advantages of the new approach, i.e. separation of the branches according to the location of maximum diabatic heating, is that the composites are clearer with respect to the feature frequencies. However, results are qualitatively similar to our original approach. The disadvantage of the new approach is that we introduce a new sensitivity, namely towards the separating longitude between the two branches. In our original approach, the longitude border can be identified more easily.

Therefore, we did not change the approach in our manuscript. However, we thank the reviewer for the undoubtedly useful suggestion.

[Figure]

**Figure R3:** *Trajectory-centred composites around location of maximum diabatic heating for the remote (old: western) and nearby (old: eastern) branch reaching upper-tropospheric anticyclones above Central Europe. Colour definitions are the same as in the manuscript.*

[Figure]

**Figure R4:** *Relative contribution of the two diabatic heating branches during the heat wave life cycle in Central Europe. The left plot shows results for 18°W as border between the remote and nearby branch, the right one for 0°E. Colour definitions are the same as in the manuscript.*

3. The novelty of this study against related work is not explicitly emphasized enough. In introduction, readers might want to know the deficiencies in related works that you will be solving, and in what way you might expect your results to differ from theirs. As mentioned in my summary: I view this study is novel in targeting heat-wave associated upper-tropospheric anticyclones. These anticyclones often collocate with heat waves [Roethlisberger et al. 2016, https://doi.org/10.1002/2016GL070944; Brunner et al. 2018, https://doi.org/10.1029/2018GL077837; Chan et al. 2019, https://doi.org/10.1029/2019GL083307], and are therefore continental anticyclones. Line 57: Instead of throwing out all the key words, you can emphasize on heat wave anticyclone, saying that they often can be weaker summertime continental anticyclones and therefore may differ from global studies like Pfahl et al. (2015) and Steinfeld and Pfahl (2019), analyses in which might be dominated by more frequent oceanic blocking. Line 60: Do you expect this study to be different and better than Quinting and Reeder (2017)? If so, please explicitly tell the difference.

Line 145: Again, lack of detailed studies of continental blocking could be the reason why this eastern branch is not known before. Could emphasize on that.

Thank you for the suggestions, we agree with your assessment. We emphasized the novelty of this study more explicitly in the introduction and added the suggested references.

The following aspects were added in the manuscript:

Regarding your comments on lines 57 and 60:
- *"Recent climatological studies on blocking tend to be dominated by oceanic blocking (Pfahl et al., 2015; Steinfeld and Pfahl, 2019), but heat waves are typically associated with summertime continental blocks (Röthlisberger et al., 2016, Brunner et al., 2018, Chan et al., 2019), which are typically weaker than wintertime blocks (Pfahl and Wernli, 2012). Also, the influence of latent heating on the formation of continental blocking may differ. Quinting and Reeder (2017) analysed trajectories reaching the lower and upper troposphere during heat waves over southeastern Australia. They emphasised the influence of cloud-diabatic processes over a baroclinic zone to the south of the Australian continent on the formation of upper-tropospheric anticyclones.*

  *However, Quinting and Reeder (2017) did not analyse the life cycle of upper-tropospheric anticyclones, i.e. whether the role of diabatic heating differs between the formation and maintenance of these anticyclones. Since Quinting and Reeder (2017) focused on Australia and no similar study exists for Europe, we therefore aim to analyse the role of diabatic heating for the formation and maintenance of upper-tropospheric anticyclones associated with heat waves in different parts of Europe."*

Regarding your comment on line 145 we added the remark that recent studies emphasized on oceanic blocks.

Minor comments
You don't have to, but I personally find the naming of western/eastern branch not intuitive enough. Is there a better alternative?
With respect to your comment 2, we changed the names of the western branch to "remote branch" and the eastern branch to "nearby branch".

The separation line of 30W is not repeatedly mentioned enough. Line 217: You might want to repeatedly remind readers that eastern means east of 30W and western means west of 30W, in this line and many other lines. Line 327: You might want to repeat in conclusions that 30W divides the two branches.

Thank you, we now repeat the separation at 30°W more often.

Line 9: "located southwest of the anticyclone" and "above western North Atlantic" are not mutually exclusive, consider saying "is located *over Africa/Europe* to the southwest. . ."

Thank you for the suggestion. We changed it to "is located over northwestern Africa/Europe to the southwest ...".

Line 22: Warming being "not spatially uniform" doesn't seem to connect well with the idea of changes in "regional circulation patterns".

We changed the sentence and only mentioned the changes in regional circulation patterns.

Line 26: This paragraph can start with a better topic sentence, saying that heat waves are associated with either an upper-tropospheric ridge or a blocking flow pattern.

Thank you for the suggestion, we slightly shortened the paragraph.

Line 27-28: In introduction, probably you don't need to include the fine details of methods in previous research.

This is correct, we have left out the details.

Line 82: Would be good to exemplify upfront that for Central Europe, 72? heat waves lasted for at least 3 days are identified.

We added that 73 heat waves were identified for Central Europe.

Line 87: You might want to explicitly mention that your definition of upper-tropospheric anticyclone requires no temporal persistence (this is implied in line 122).

We now mention this in our manuscript.

Line 109: Might be good to be slightly clearer about the difference in method to Steinfeld and Pfahl (2019).

The main difference to Steinfeld and Pfahl (2019) is that in their study *all* trajectories experiencing diabatic heating of more than 2 K are categorised as "diabatically heated", no matter how large the cooling is. This is mentioned in the revised manuscript.

Line 233: Might be useful to show the figure for the pressure of maximum diabatic heating.

We added a new figure showing pressure and timestep of maximum diabatic heating (the latter one was a supplementary figure before).

Line 270: Is the idea of Quinting and Reeder (2017) more like warm conveyor belts in the western branch? Or more like the eastern branch? Could be more explicit.

The idea is more like warm conveyor belts in the remote (old: western) branch. We added this in the revised version.

Figure captions: Proofreading or copy-editing is needed. (Plurals, Capital letters,

spaces, etc.)
Figure 7b caption line 6: Please note in caption that orange shading in 7b is not visible.

We deleted the sentence mentioning the orange shading.

Figure 8 caption: Please note in caption that WCB is not visible.

We added a comment on this to the caption.

Figure 8 purple hatching: Do you require total precipitation >= 2 mm/d? You might also want to remind readers that purple hatching in Fig. 8 is opposite to that in Fig. 7.

We completed the caption.

Technical corrections

Line 162: is found east of the *western* heating branch.

Corrected.

Line 224: over the *European* continent. . .

Corrected.

Line 236: *42* to 54 h

Corrected. Thank you for the careful reading.

Line 296: "About 70" -> "72"?

73 heat waves, corrected.

Line 297: a duration of *at least* three days

Corrected.

Line 332: Are there *three* source regions instead of two?

No, only two. The nearby heating branch and the cooling branch originate from a similar region (although from different pressure levels, but we would not see this as a different source region). We added the word *geographic* source regions to make this clear.

Line 376: Do you mean Rossby wave *packets*?

Yes, corrected accordingly.

Figure 3 caption line 4: boxes represents -> boxes represent

Corrected.

Figure 6 caption line 3: ... and Greece/ Italy (GI).

Added.

Figure 7 caption line 6: warm conveyor belts *(red)* frequency

Corrected.

Figure 8 caption line 2: grey shading -> purple hatching?

Corrected.

---

## Author Response (AR2)

**Replies to the reviewer of "A Lagrangian analysis of upper-tropospheric anticyclones associated with heat waves in Europe" by Philipp Zschenderlein et al.**

We thank the reviewer for his/her interest and the time he/she spent to review our manuscript a second time. The comments of the reviewer is given in black, our replies in green colour.

This is my second review of this manuscript.

As already stated in my comment of the original submission, I find this work scientifically significant and methodologically sound. I also find that in the revised manuscript and the response letter, the authors covered my comments in a way more than satisfactory.

After careful evaluation I can therefore recommend the manuscript for publication.

Minor comments (line numbers in author response)

Line 60-61: weaker than wintertime *oceanic* blocks.

Added to the revised manuscript.

Line 97: duration of the surface heat waves *(at least 3 days)*.

Added.

Line 338: "About 73" -> "73"

Corrected.

Figure 4 caption: "but seven days prior to arrival."

Added.

Figure S3: Something go beyond the northern side of the bounding box. Please adjust map limits.

We produced new plots to include the northern side of the bounding boxes.

Figure S4 caption line 6: warm conveyor belt *(red)* frequency

Added.

[revised manuscript text omitted]